# Improving the Agronomic Value of Paddy Straw Using *Trichoderma harzianum, Eisenia fetida* and Cow Dung

**Neetu Sharma** [1], **Jagjeet Singh** [1], **Bijender Singh** [2,3,*] **and Vinay Malik** [1,*]

1   Department of Zoology, Maharshi Dayanand University, Rohtak 124001, India; neetsjangra@gmail.com (N.S.); jagjeet.rs.zoo@mdurohtak.ac.in (J.S.)
2   Department of Biotechnology, Central University of Haryana, Jant-Pali, Mahendergarh 123031, India
3   Laboratory of Bioprocess Technology, Department of Microbiology, Maharshi Dayanand University, Rohtak 124001, India
*   Correspondence: ohlanbs@gmail.com (B.S.); vinaymalikzoo@mdurohtak.ac.in (V.M.)

**Abstract:** The aim of the present study was to assess the effects of inoculation of *Trichoderma harzianum*, *Eisenia fetida* and cow dung on the physicochemical quality of paddy straw composting which was carried out for 90 days. The different treatment groups were Paddy straw ($T_0$), Paddy straw + Cow dung ($T_1$), Paddy straw + Cow dung + *Eisenia fetida* ($T_2$), Paddy straw + Cow dung + *Trichoderma harzianum* ($T_3$), Paddy straw + Cow dung + *Eisenia fetida* + *Trichoderma harzianum* ($T_4$). The ratio of cow dung and paddy straw was 2:1. Among all treatments, $T_4$ was identified as the best treatment for decomposing the paddy straw as it recovered the nutrients within the recommended levels of a high-quality product. The consortium of *Trichoderma harzianum*, *Eisenia fetida* and cow dung lowered the total organic carbon (TOC) and C:N ratio by 28.8% and 33.1%, respectively, at pH 6.5. The increase in N (0.87%), P (0.47%), K (2.66%), Ca (0.033%), Mg (0.056%) and Na (0.42%) was significant in $T_4$ treatment. The micronutrients, namely Cu (47.9 ppm), Fe (1128 ppm) and Zn (500 ppm), also showed a significant increase in this treatment, i.e., $T_4$. Therefore, results suggested that combinatorial composting by *Trichoderma harzianum*, *Eisenia fetida* and cow dung is quite promising in the decomposition of paddy straw to obtain quality compost in a short time. Furthermore, this study will help in the sustainable management of paddy straw with concomitant reduction inenvironmental pollution caused by the open burning of paddy straw.

**Keywords:** paddy straw; composting; *Eisenia fetida*; *Trichoderma harzianum*; micronutrients

## 1. Introduction

Approximately 715 million tons of paddy is produced annually in the world, of which India and China alone account for 50% production [1]. Paddy straw production worldwide is 760 million tons, which is 1.5 times greater than paddy production [2,3]. In Asian countries, India is the second leading producer of paddy producing 152.5 million tons of paddy and 157.2–235.8 million tons of paddy straw yearly [2–5]. Management of this massive amount of crop residue is a concern across the world as farmers cannot wait for its degradation to sow the next crop. Therefore, 85–95 million tons of paddy straw in India is burnt in open fields because it is neither easily degraded nor palatable due to its high lignocellulosic and silica contents. High collection costs, unavailability of economic benefits to utilize the straw and less time between harvesting rice and sowing wheat, constrained farmers to burn rice straw in the fields [6]. Repeated burning cause soil erosion and environmental pollution, which adversely affects public health [7].

The paddy straw is a typical lignocellulosic waste containing 30–45% cellulose, 20–25% hemicellulose and 15–20% lignin with a small number of organic compounds. The high C:N ratio of rice straw makes it a less biodegradable waste in comparison to other agricultural wastes [8].This is the reason for the slow degradation of its organic constituents. Composting is one of the eco-friendly biological processes for recycling nutrients in agricultural

fields. It is a transformation process that changes agricultural wastes into a biodegradable stabilized and mineralized humus by microorganisms including bacteria and fungi [9], while vermicomposting is a process that converts biodegradable waste into organic manure with the help of mainly earthworms and microorganisms. Vermicomposting is a biochemical process that recycles nutrients in the soil and makes them available for plant growth. Due to the presence of microbes, it is considered a high-nutrient fertilizer with a large microbial community [10]. The generated compost can increase or replenish the organic matter in agricultural fields by decreasing the need for chemical fertilizers in the crops. The compost, which is a rich source of mineralized nutrients, not only improves the soil's health but also improves crop productivity [11].

Composting along with vermicomposting are among the most common practices for recycling paddy straw. Since both are low-cost technologies for converting paddy straw into a nutrient-rich product, making them a promising strategy for managing this agricultural waste [10].The degradation of lignocellulosic wastes by conventional composting is a slow and time-consuming process. At the same time, a combination of appropriate, effective bacteria, actinomycetes or a suitable group of fungi can accelerate the decomposition. The inoculum of lignocellulolytic microorganisms may be a good substitute for the in situ burning of paddy straw to produce compost in a more economical manner in less time [12]. A consortium of *Trichoderma viride* and *Aspergillus niger* can efficiently decompose hemicelluloses, lignin, cellulose and total carbon of rice straw [13]. The most common bacterial species involved in the degradation of paddy straw are *Staphylococcus*, *Bacillus* and *Klebsiella* [14], whereas fungal species involved in the degradation of paddy straw are *Aspergillus*, *Fusarium* and *Trichoderma*,in which *Trichoderma* species are among the most effective fungi for decomposing the lignocellulosic waste [15]. Sitepu et al. [16] investigated the composting of paddy straw by *Aspergillus niger* and *Trichoderma viride*. *Trichoderma harzianum* using its well-balanced cellulolytic complex hydrolyzes the cellulosic fraction of plant biomass into monomeric glucose. Due to its efficient cellulolytic activity, *T. harzianum* has significant potential in biomass hydrolysis which may be used in paddy straw degradation [17].

The consortium of earthworms and lignocellulolytic microbiota plays an important role in solid waste degradation, as they decrease the composting period by altering the structure of solid waste with their enzymatic actions [9,18]. Microbial communities and enzyme activities can be used to describe the dynamics of the composting process, including the decomposition of organic matter, its stability and its maturity [19]. Earthworms alter both the biological and physical properties of organic matter [20]. Earthworms have a strong potential for regulating straw decomposition and their inoculation significantly accelerated paddy straw degradation and promoted the conversion of straw carbon into soil carbon [21,22]. The excreta of earthworms are rich in bacteria and enzymes thatare contributed by the secretion of mucopolysaccharides of earthworm's intestine and skin [23]. Epigeic earthworms including *Eisenia fetida*, *Lumbricus castaneus*, *L. rubellus*, *L. festivus*, *Eiseniella tetraedra*, *Bimastus eisenia* and *B. minusculus* have high reproduction rates, a short life cycle, active gizzard and habitat tolerance [10]. Jaybhaye and Bhalerao [24] reported that *Eisenia fetida* has a high capability to feed on organic matter and ingest five times its body weight. Organic matter is biochemically degraded with the help of microbes while the conditioning of organic matter is carriedby earthworms. Cow dung is beneficial in sustainable development as a bioresource [25]. It was utilized as a microbial inoculum due to the presence of fungi, bacteria and protozoa [26]. So, the aim of this study was to decompose the paddy straw with *Trichoderma harzianum*, *Eisenia fetida* and cow dung for its sustainable management and enhancing its agronomic value by enhancing the nutrients in the compost.

## 2. Material and Methods

### 2.1. Collection of Paddy Straw, Eisenia fetida, and Trichoderma harzianum

Fresh paddy straw was collected from regional fields after harvesting of the crop and was chopped approximately 1–2 cm and placed in trays (290 × 220 × 140 mm). Five hundred earthworms *Eisenia fetida* (epigeic species) were procured from the Department of Agronomy, Indian Agricultural Research Institute (IARI), New Delhi, and cultured in laboratory for 120 days by using old cow dung as feeding material. Temperature (25–30 °C) and moisture (60–70%) of bedding were maintained for proper growth of *Eisenia fetida*. The fungal culture *Trichoderma harzianum* (ITCC 6721) was procured from the Indian type culture collection (ITCC), IARI, New Delhi. It was cultured in the laboratory by preparing media of potato dextrose agar. After autoclaving, all the media weretransferred into culture plates and placed for 24 h. *Trichoderma harzianum* was inoculated in culture plates in laminar airflow. After inoculation, culture plates were incubated at 28 °C for growth. The spore suspension was prepared by extracting the spores from the culture plates with the help of normal saline (0.85% NaCl and 0.5% Tween 80).

### 2.2. Experimental Design with Treatment Conditions

Composting process was carried out with five different experimental set up, which were Paddy straw ($T_0$), Paddy straw + Cow dung ($T_1$), Paddy straw + Cow dung + *Eisenia fetida* ($T_2$), Paddy straw + Cow dung + *Trichoderma harzianum* ($146.1 \times 10^7$ spores) ($T_3$), Paddy straw + Cow dung + *Eisenia fetida* + *Trichoderma harzianum* ($146.1 \times 10^7$ spores) ($T_4$), wherein paddy straw and cow dung ratio was 2:1. All treatments were pre-composted for a week followed by inoculation of 30 adult *Eisenia fetida* (with well-developed clitellum) in treatment $T_2$ and $T_4$ and *Trichoderma harzianum* ($146.1 \times 10^7$ spores) in $T_3$ and $T_4$,which lasted for 90 days. Each treatment was replicated thrice and water was sprinkled to maintain the moisture (60–70%) and covered with muslin cloth for proper aeration throughout composting. The samples were taken out at regular intervals and analyzed for different parameters.

### 2.3. Physico-Chemical Analysis

The compost samples were dried at room temperature and sieved with a 2 mm sieve and used for physicochemical analysis.

The pH of compost was recorded by taking a 2 g sample of compost in 10 mL of distilled water (1:5 ratio) and stirring it for 20 min.The solution mixture was recorded using a calibrated digital pH meter according to Jodice et al. [27].

Total organic carbon (TOC) was measured by taking 1 g of sample in a China crucible and placedinto an electric furnace for 24 h at 700 °C. After that, the electric furnace was permitted to cool and the ash produced was weighed according to the dry combustion method of Nelsen and Sommers [28].

Total nitrogen (N) content was measured by using the Kjeldahl method after digesting a 0.5 g sample of dry compost in concentrated $H_2SO_4$ mixture and $HClO_4$ (ratio 1:9) [29].

Potassium (K) and sodium (Na) content were measured through flame photometry after digesting a 0.5 g sample of dry compost in concentrated $HNO_3$ mixture and $HClO_4$ (ratio 4:1) [30].

The titration method was used for the determination of magnesium (Mg) and calcium (Ca) by using ammonium acetate with standard EDTA solution (1:5) and Eriochrome Black T indicator [31].

The vanadomolybdate method was used for the estimation of phosphorus (P) according to Koenig and Johnson [32].

Micronutrients like copper (Cu), zinc (Zn) and iron (Fe) were measured by using atomic absorption spectrophotometer (Perkin Elmer, Waltham, MA, USA, AAnalyst 100) after digesting a 0.5 g sample in 10 mL of diacid mixture of concentrated nitric acid ($HNO_3$) and perchloric acid ($HClO_4$) in ratio of 1:4 [33].

*2.4. Statistical Analysis*

All data with three replicates for different nutrient concentrations were examined by applying ANOVA (two-way analysis of variance) to compare the average value with post hoc Tukey's HSD test for significance according to the experimental design in SPSS software (version 25.0). *p* values < 0.05 represents statistical significance for all analysis. GraphPad Prism version 0.2 was used to identify the differences in the concentration of nutrients during the composting process.

**3. Results and Discussion**

The different physicochemical qualities of the mature compost were analyzed in the present study. The processed material was more stabilized, odor-free and nutrient-rich than the initial material of paddy straw. The results of paddy straw composting reporting enriched nutrient contents are shown in Tables 1 and 2 and Figures 1–3.

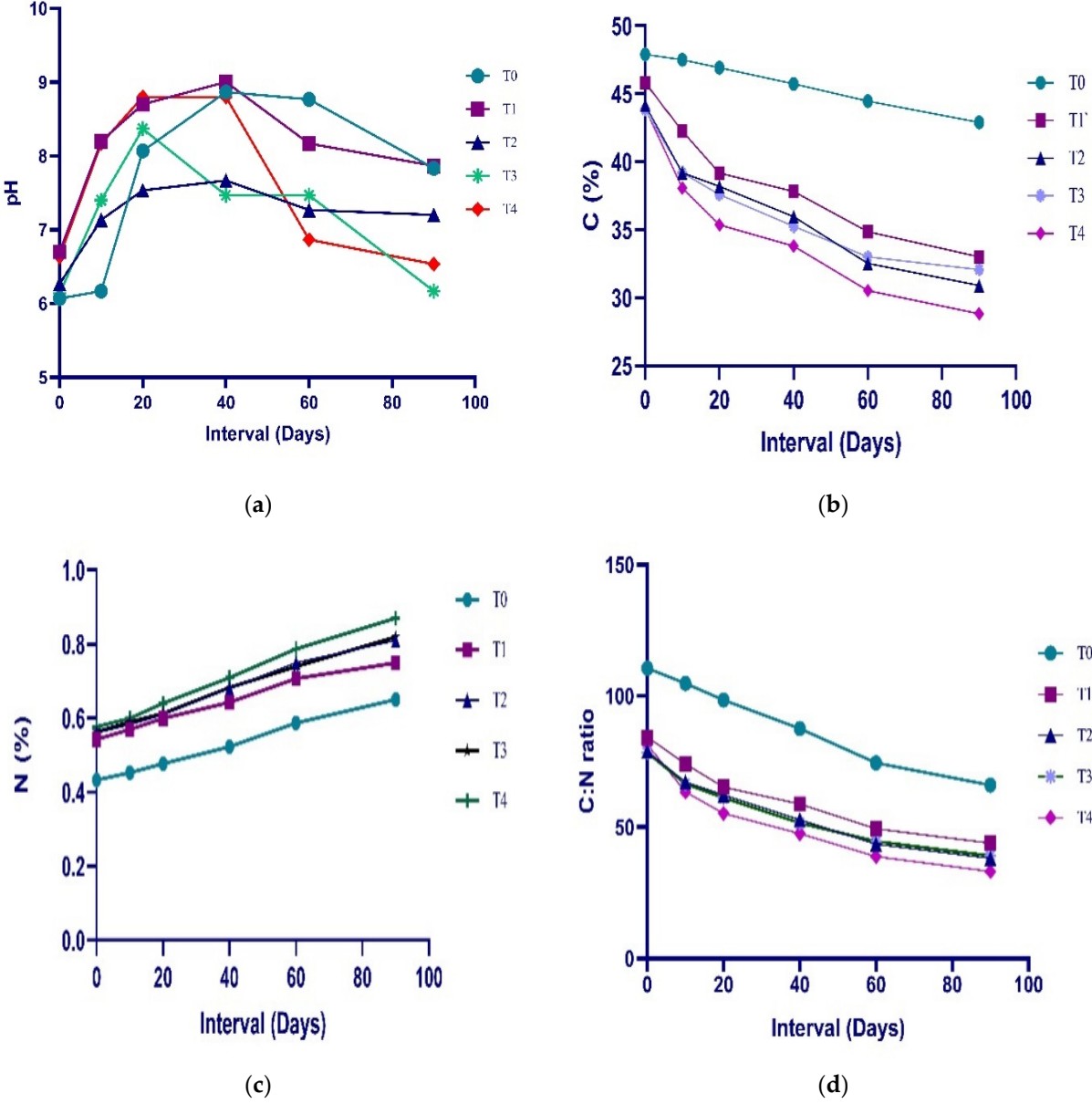

**Figure 1.** Change in (**a**) pH, (**b**) total organic carbon (TOC%), (**c**) nitrogen (N%) and (**d**) C:N ratio during paddy straw composting in different treatments $T_0$, $T_1$, $T_2$, $T_3$, $T_4$. Values are represented as Mean ± SE. *n* = 3.

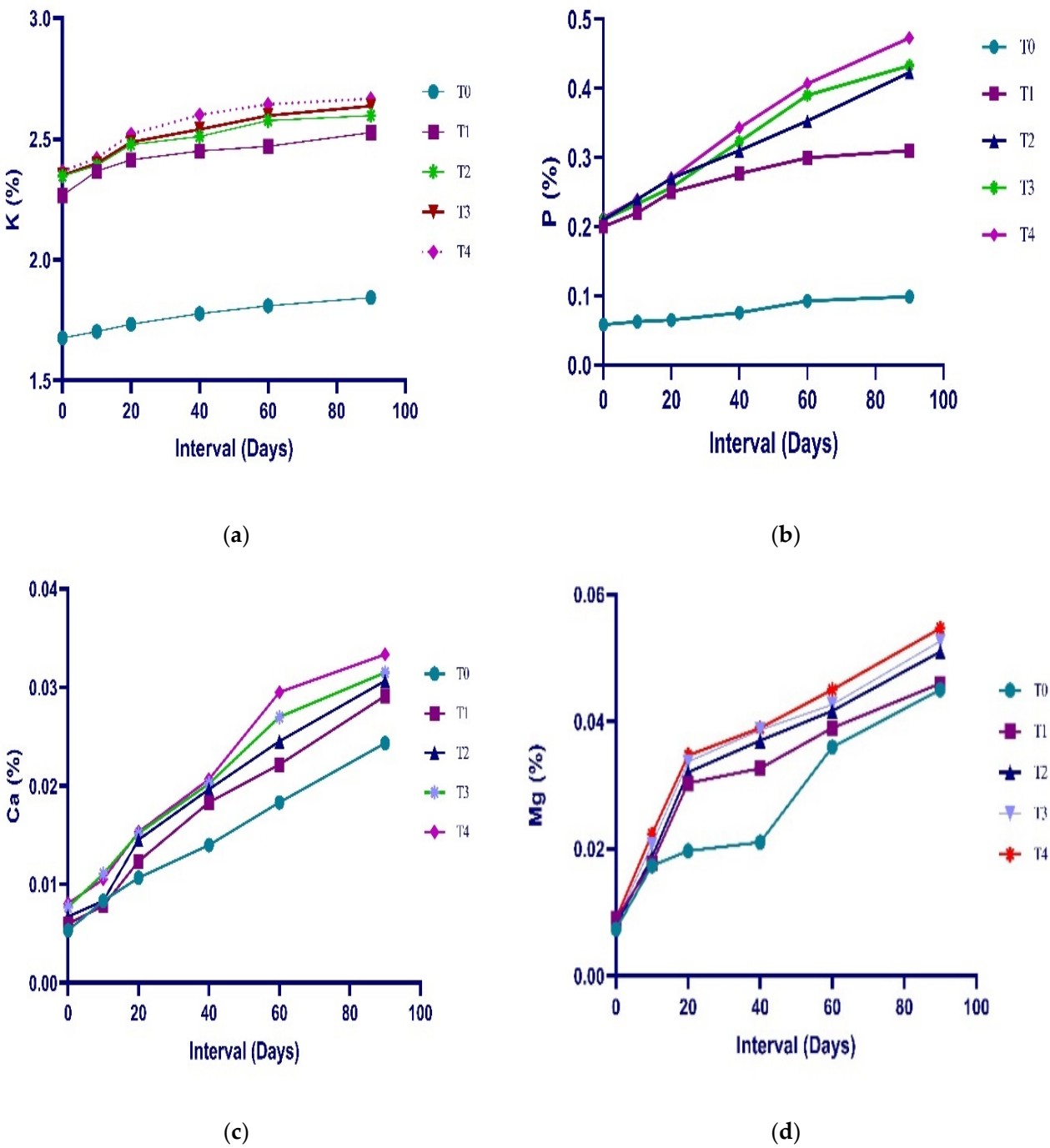

(**a**)

(**b**)

(**c**)

(**d**)

**Figure 2.** Change in (**a**) potassium (K%), (**b**) phosphorous (P%), (**c**) calcium (Ca%) and (**d**) magnesium (Mg%) during paddy straw composting in different treatments $T_0$, $T_1$, $T_2$, $T_3$, $T_4$. Values are represented as Mean $\pm$ SE. $n = 3$.

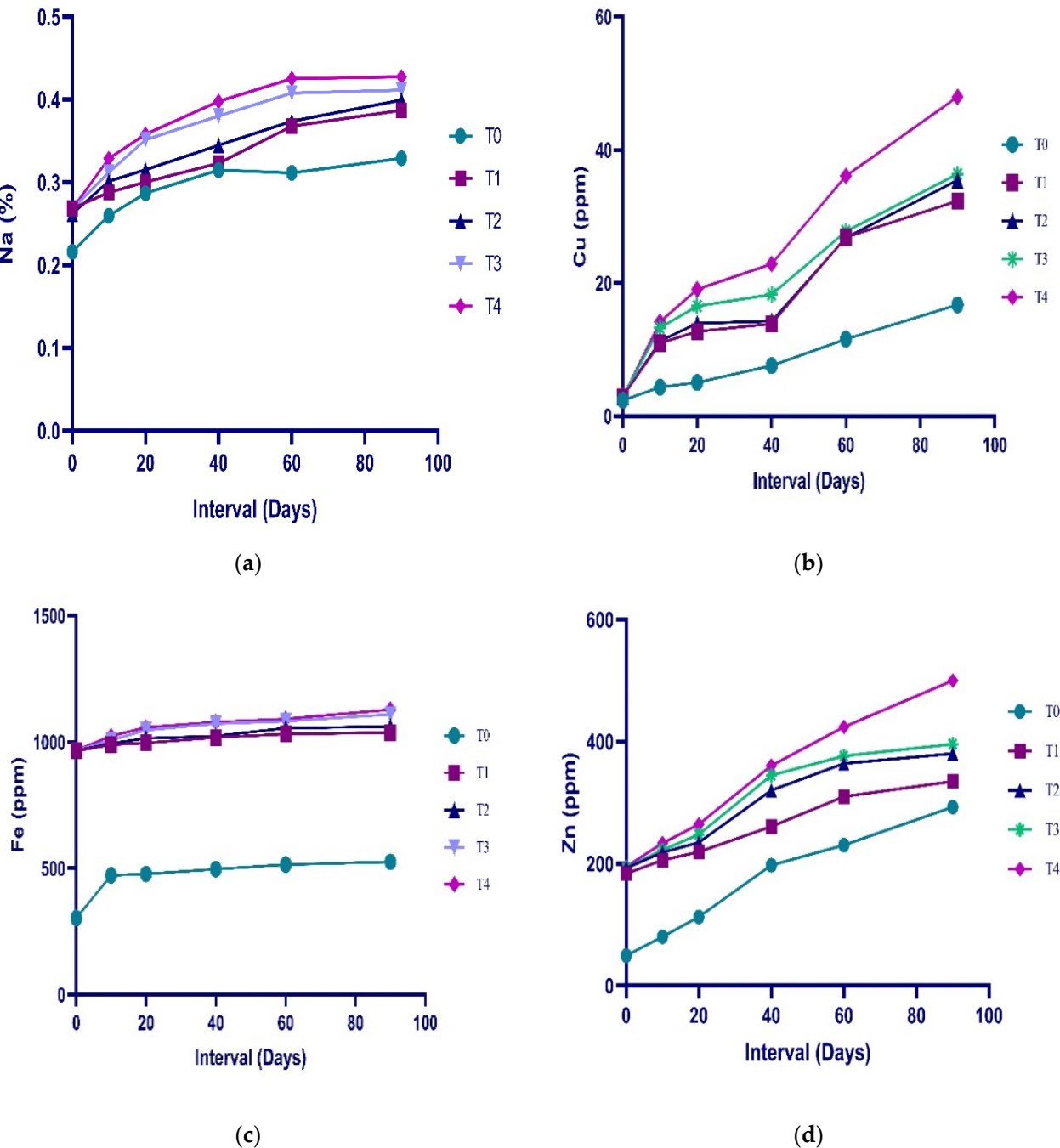

**Figure 3.** Change in (**a**) sodium (Na%) (**b**) copper (Cu in ppm), (**c**) iron (Fe in ppm) and (**d**) zinc (Zn in ppm) during paddy straw composting in different treatments $T_0$, $T_1$, $T_2$, $T_3$, $T_4$. Values are represented as Mean ± SE. *n* = 3.

**Table 1.** pH, carbon, nitrogen, C:N ratio, potassium and phosphorous at the start and after paddy straw composting of different treatments. Values are represented as Mean ± SE. *n* = 3. Asterisks (*) represented significance at ($p < 0.05$) as compared to control based on Tukey's HSD test. Values with superscript (a) in carbon, nitrogen and C:N ratio in $T_2$ and $T_3$ are non-significant with each other. Values in parenthesis are percent increase/decrease over control.

| Treatment | Interval (Days) | | | | | | | | | | | |
|---|---|---|---|---|---|---|---|---|---|---|---|---|
| | pH Value | | Carbon (%) | | Nitrogen (%) | | C:N Ratio | | Potassium (%) | | Phosphorous (%) | |
| | At Start | At End | At Start | At End | At Start | At End | At Start | At End | At Start | At End | At Start | At End |
| $T_0$ | 6.06 | 7.83 * (29.21) | 47.90 ± 0.40 | 42.90 ± 0.14 * (10.44) | 0.43 ± 0.002 | 0.65 ± 0.004 * (51.16) | 110.50 ± 1.52 | 66.01 ± 0.27 * (40.26) | 1.67 ± 0.007 | 1.84 ± 0.004 * (10.17) | 0.05 ± 0.01 | 0.09 ± 0.002 * (80) |
| $T_1$ | 6.7 | 7.86 * (17.31) | 45.83 ± 0.14 | 33 ± 0.44 * (27.99) | 0.54 ± 0.002 | 0.75 ± 0.004 * (38.89) | 84.30 ± 0.44 | 43.93 ± 0.33 * (47.89) | 2.26 ± 0.004 | 2.52 ± 0.004 * (11.50) | 0.2 ± 0.009 | 0.31 ± 0.01 * (55) |
| $T_2$ | 6.26 | 7.2 * (15.02) | 44.16 ± 0.42 | 30.90 ± 0.41 [a]* (30.03) | 0.56 ± 0.004 | 0.81 ± 0.004 [a]* (44.64) | 78.86 ± 1.20 | 38.13 ± 0.61 [a]* (51.65) | 2.34 ± 0.004 | 2.59 ± 0.01 * (10.68) | 0.21 ± 0.02 | 0.42 ± 0.02 * (100) |
| $T_3$ | 6.13 | 6.16 * (0.49) | 43.83 ± 0.22 | 32.06 ± 0.64 [a]* (26.85) | 0.56 ± 0.007 | 0.82 ± 0.004 [a]* (46.43) | 78.43 ± 1.79 | 39.06 ± 0.84 [a]* (50.20) | 2.35 ± 0.004 | 2.63 ± 0.007 * (11.91) | 0.21 ± 0.01 | 0.43 ± 0.03 * (104.7) |
| $T_4$ | 6.63 | 6.5 * (1.96) | 43.93 ± 0.31 | 28.83 ± 0.26 * (34.37) | 0.57 ± 0.007 | 0.87 ± 0.004 * (52.63) | 81.53 ± 2.61 | 33.10 ± 0.16 * (59.40) | 2.36 ± 0.002 | 2.66 ± 0.007 * (12.71) | 0.21 ± 0.01 | 0.47 ± 0.02 * (123.8) |

**Table 2.** Calcium, magnesium, sodium, copper, iron and zinc at start and after paddy straw composting of different treatments. Values are represented as Mean ± SE. *n* = 3. Asterisks (*) represented significance at ($p < 0.05$) as compared to control based on post hoc Tukey's HSD test. Values in parenthesis are percent increase/decrease over control.

| Treatment | Interval (Days) | | | | | | | | | | | |
|---|---|---|---|---|---|---|---|---|---|---|---|---|
| | Calcium (%) | | Magnesium (%) | | Sodium (%) | | Copper (ppm) | | Iron (ppm) | | Zinc (ppm) | |
| | At Start | At End | At Start | At End | At Start | At End | At Start | At End | At Start | At End | At Start | At End |
| $T_0$ | 0.005 ± 0.0005 | 0.024 ± 0.0007 * (380) | 0.007 ± 0.0007 | 0.045 ± 0.0009 * (542) | 0.21 ± 0.006 | 0.32 ± 0.003 * (53.2) | 2.38 ± 0.005 | 16.73 ± 0.66 * (602) | 302 ± 1.72 | 525 ± 4.47 * (73.8) | 49.25 ± 0.16 | 292 ± 4.00 * (492) |
| $T_1$ | 0.006 ± 0.0005 | 0.029 ± 0.0005 * (383) | 0.009 ± 0.0004 | 0.046 ± 0.0004 * (411) | 0.26 ± 0.003 | 0.38 ± 0.004 * (46.4) | 2.90 ± 0.002 | 32.33 ± 0.61 * (1014) | 964 ± 3.65 | 1036 ± 2.27 * (7.46) | 183.4 ± 1.43 | 334 ± 2.23 * (82.5) |
| $T_2$ | 0.006 ± 0.0002 | 0.030 ± 0.0007 * (400) | 0.008 ± 0.0004 | 0.051 ± 0.001 * (537) | 0.26 ± 0.004 | 0.39 ± 0.004 * (50) | 2.90 ± 0.09 | 35.44 ± 0.58 * (1122) | 965 ± 2.63 | 1061 ± 2.34 * (9.94) | 192.8 ± 1.01 | 380 ± 2.94 * (97.09) |
| $T_3$ | 0.007 ± 0.0002 | 0.031 ± 0.0002 * (342) | 0.008 ± 0.0005 | 0.053 ± 0.001 * (562) | 0.26 ± 0.004 | 0.41 ± 0.004 * (47.6) | 2.93 ± 0.01 | 36.36 ± 0.60 * (1138) | 967 ± 3.77 | 1109 ± 3.90 * (14.68) | 193.4 ± 1.20 | 395 ± 2.47 * (104.23) |
| $T_4$ | 0.008 ± 0.0005 | 0.033 ± 0.0007 * (312) | 0.009 ± 0.0004 | 0.056 ± 0.001 * (522) | 0.26 ± 0.002 | 0.42 ± 0.003 * (61.5) | 2.85 ± 0.02 | 47.93 ± 0.28 * (1581) | 967 ± 2.75 | 1128 ± 3.85 * (16.64) | 194.9 ± 2.33 | 500 ± 4.61 * (156.54) |

### 3.1. Analysis of pH

The pH values of the five treatments were in the ranges of 6.06 ($T_0$)–9.0 ($T_1$) during composting (Figure 1a). A trend of initial increase was followed by a gradual decrease in all the treatments. The treatment $T_3$ (6.16) and $T_4$ (6.5) were slightly acidic till the end of composting (Table 1). The change in pH could be due to the production of organic acids and other intermediate compounds (e.g., $CO_2$, $NH_3$, $NO_3^-$) by the action of microorganisms and earthworms [34]. The high pH values in the initial period (up to 40 days) may be due to the mineralization of organic nitrogen into ammonia. The decrease in pH conserves the nitrogen in the final compost as nitrogen is lost as volatile ammonia. The compost's pH determines the efficiency of composting by affecting the nutrient's availability. Microbial activity is also affected by the pH of composting as they grow well in neutral pH [35]. Thus, the rate of production of organic acids and other intermediate compounds is affected by pH changes which consequently affects the degree of mineralization and microbial population in the composting material [34]. The results were comparable to the findings of other studies [36–40].

### 3.2. Analysis of Total Organic Carbon (TOC)

The TOC decreased significantly in all the treatments, and the changes in TOC during composting are shown in Figure 1b. The rate of TOC loss in treatments was in the range of 43.8 ($T_3$)–47.9 ($T_0$) at the start, whereas at the end of composting, it was 28.8 ($T_4$)–42.9 ($T_0$) which is a decrease of 0.6–0.9 folds. The TOC loss in treatments was in the order of $T_4$ (34.37%) > $T_2$ (30.03%) > $T_1$ (27.99%) > $T_3$ (26.85%) > $T_0$ (10.44%) [Table 1]. A significant difference was reported between treatments $T_0$, $T_1$, $T_2$, $T_3$ and $T_4$, but there was a non-significant difference between $T_2$ and $T_3$ treatments according to Tukey's test ($p < 0.05$). The results showed a similar trend of TOC loss in all treatments except $T_0$. The combination of *E. fetida*, *T. harzianum* and cow dung enhanced carbon mineralization at the fastest rate. The loss in TOC may be due to microbial respiration during composting [41]. Further digestion and assimilation of carbon from paddy straw by earthworms cause carbon loss from the composting material. The longitudinal folds of the intestinal wall in the intestine of earthworms increase the surface area for microbial degradation of agricultural wastes [42]. Moreover, the combined action of gut microbes and earthworms increased carbon mineralization, leading to a loss of mass in the paddy straw mixture, while cellulolytic enzymes produced by fungi also lead to carbon loss by breaking down the cellulose into glucose [43]. Sharma and Garg. [30] reported a 17.38–58.04% decrease in TOC after composting rice straw. Arora and Kaur. [6] reported a 4.58–8.02% decrease in TOC in the mixture of rice straw, *Azolla*, cattle dung and earthworms. The results of the reduction in TOC are also in accordance with similar studies of composting [23,41,42,44,45].

### 3.3. Analysis of Nitrogen (N)

There was an increase in the N contents in all the treatment groups at the end of the experiment. The N in prepared compost was in the range of 0.65 ± 0.004–0.87 ± 0.004. The maximum increase in N content was observed in $T_4$ (52.63%) followed by $T_0$ (51.16%), $T_3$ (46.43%), $T_2$ (44.64%) and $T_1$ (38.39%) (Table 1). A statistically significant difference was shown within the treatments except for $T_2$ and $T_3$ ($p < 0.005$). A trend of steady, gradual increase was recorded in all the treatment groups (Figure 1c). The N mineralization was related to the mixture contents of the treatment groups. Data suggested that the N increase was different in different degrading inoculations. Earthworms changed the microbiota responsible for N enrichment during composting, probably due to their excretory products, carbon rich mucus secretions, body fluids and enzymes [46], while fungi provide extracellular enzymes like cellulases, xylanases and laccases, which degraded cellulose, hemicellulose and lignin components of rice straw. Moreover, continuous earthworm movement caused high aeration leading to aerobic conditions, consequently minimizing the N loss during nitrification [43]. Results, thus, indicated better enrichment of N in the presence of *E. fetida*. This indicated the suitability of combinatorial composting in the

degradation of paddy straw wastes. Several earlier studies also reported an increase of 0.9 to 1.5-fold in N contents during composting [23,43,46–48].

### 3.4. Analysis of C:N Ratio

In this study, the initial C:N ratios were in the range of 78.43–110.5, as shown in Table 1. The C:N ratio of the finished compost of all the tested treatments range from 33 to 66. The study reported a decrease of 0.4–0.6-fold from the start of composting depicting an overall decrease in C:N ratio of all the treatments, as shown in Figure 1d. The maximum reduction in C: N was found in $T_4$ (59.4%), followed by $T_2$ (51.6%), $T_3$ (50.2%) and $T_1$ (47.8%). The treatments $T_2$ and $T_3$ recorded a non-significant decrease being 38.1 and 39, respectively. The treatment $T_4$ showed a maximum decrease in C:N ratio (59.4%) with a significant difference ($p < 0.05$) from the other treatments indicating fast mineralization due to loss of carbon and nitrogen enrichment in the presence of *E. fetida* and *T. harzianum* and cow dung. The C:N ratio of less than 20 is traditionally used as an indicator of the maturity of compost, defining its agronomic value. Results, thus, indicate that C:N ratio of matured compost of $T_4$ was higher than the recommended range of good compost as reported by Pandit et al. [49]. Still, the C:N ratio of nearly 30 of the $T_4$ treatment of paddy straw, which is a high lignocellulose content substrate reflects a satisfactory degree of maturity of the finished compost in the present study. However, some studies disagree with the correlation between C:N ratio and compost maturity, as C:N ratio varies greatly with substrate material. Thus, it may be a misleading indicator of the maturity of the compost and may not suggest a sufficiently decomposed substrate [50].

### 3.5. Analysis of Potassium (K)

The K content was 1.1 to 1.13-fold higher in the prepared compost than in the initial paddy straw mixture. In mature compost, K content was in the range of 1.67–2.66% (Table 1). The K changes in the paddy straw mixtures were relative, showing a gradual increase, except for $T_0$, which showed very little increase during the experiment. A static trend in K content was recorded after 40 days (Figure 2a). All treatments showed significant differences in K ($p < 0.05$). K is an essential element for plant growth, its high level in ready compost indicates the suitability of prepared compost for crop growth in fields. The various endogenic and exogenic enzymes of the gut of the earthworms convert the non-available organic minerals into available soluble forms. Fungal hyphae, with their absorptive nature, bind the available K with cellulose-rich material, consequently helping increase the K contents in the composting mixtures. The mutual action of earthworms and fungi with their enzymatic actions plays a vital role in microbial-mediated nutrient mineralization in paddy straw decomposition. These results are in line with the previous studies of lignocellulosic waste composting [23,30,43,51,52].

### 3.6. Analysis of Phosphorous (P)

The total P in the initial mixture except control $T_0$ was 0.21% and showed a significant increase with 0.47% at the end of composting ($p < 0.05$) (Table 1). The P changes in all paddy straw mixtures during the composting are shown in Figure 2b. P showed a similar trend of increase, except for $T_2$ and $T_4$, which showed a sharp increase after 20 days of composting. P, along with N and K, is an essential nutrient required for optimum plant growth. The high P level in the mature compost suggests better plant growth for sustainable crop production. The phosphate mobilization and mineralization are due to gut phosphatases of the earthworm and the additional release of P by the microbiota associated with earthworm casts. Additionally, the fungus also has the ability of P solubilization giving a suitable environment to the microbes solubilizing P during the composting process. There was an increase of 123.8% in P contents during composting in the present study, which is in accordance with other earlier studies of composting [6,23,30,43,53–55].

### 3.7. Analysis of Calcium (Ca)

Calcium acts as a structural component of the cell wall and helps to maintain the turgidity of the plant cell [56]. It is also required for cell signaling and cell wall stabilization [57]. The Ca content in the different treatment groups varied between 0.005 and 0.33% recording higher Ca content in the mature compost (Table 2). A continuous increasing trend of Ca mineralization is shown in Figure 2c. A statistically significant difference was shown within the treatments ($p < 0.005$). The Ca mineralization rate showed a steep increase in all the treatment groups after 20 days. Earthworm's gut activities are primarily responsible for the increased Ca contents in the mature compost. It may be due to earthworm's calciferous glands which are excretory organs responsible for releasing excess Ca as calcium carbonate [58]. Calcium is efficiently utilized by plants in the form of calcium carbonates [6]. Further, Ca mineralization is enhanced by the fungal hyphae besides microbiota associated with the earthworm's cast [59]. These results are in line with the previous studies of lignocellulosic waste composting [6,60,61].

### 3.8. Analysis of Magnesium (Mg)

In the present study, Mg content in all treatments at the final stage was 0.045%, 0.046%, 0.051%, 0.053 and 0.054% in $T_0$, $T_1$, $T_2$, $T_3$ and $T_4$, respectively, showing a significant difference between all the treatments ($p < 0.05$), (Figure 2d). The increasing trend of Mg was $T_0$ (257%), followed by $T_1$ (411%), $T_2$ (537%), $T_3$ (562%) and $T_4$ (600%) (Table 2). The activities of earthworms support the colonization of fungi, which increases the availability of Mg in organic matter [62]. Vermicomposting of rice straw, when mixed with cattle dung, increases Mg content (17–40.8%) significantly [63]. Arora and Kaur [6] observed a similar trend of increased Mg in rice straw vermicompost. The enhanced contents of Mg in the mature compost indicated its enhanced nutritive value.

### 3.9. Analysis of Sodium (Na)

Sodium is an important plant nutrient, as it prevents soil acidification by increasing other nutrients in the form of oxides, hydroxides and carbonates [19,64]. In the present study, Na content increased in all the treatments compared to the initial values, as illustrated in Figure 3a. The Na content in prepared compost was in the range of $0.32 \pm 0.003$ to $0.42 \pm 0.003\%$, which was 1.52–1.61 fold higher than the initial Na content (Table 2). There was a significant difference between all the treatments ($p < 0.05$). Several studies also reported an increase in the Na content of vermicompost made from various wastes [6,30,65,66].

### 3.10. Micronutrient (Cu, Fe and Zn) Profile during Composting

Micronutrients like Cu, Fe and Zn are needed in small quantities, but have a significant impact on plant growth because they act as co-factors for many antioxidant enzymes like catalase, peroxidase and superoxide dismutase [6]. The micronutrient concentrations of Cu, Fe and Zn, contents of paddy straw composting are shown in Table 2. The changes in Cu, Fe and Zn content during composting of paddy straw is shown in Figure 3b–d. There was a significant difference between all the treatments ($p < 0.05$), and it was observed that Cu, Fe and Zn content increased in all the treatments. The range of variation in Cu, Fe and Zn content was between 2.38 ($T_0$)–47.93 ($T_4$) ppm, 302 ($T_0$)–1128 ($T_4$) ppm and 49.25 ($T_0$)–500 ($T_4$) ppm, respectively. The Cu content showed a 7.02–17.1-fold increase in treatments. In comparison, the iron content was 1.07–1.73-fold higher than the initial level. The Zn content was highest in the $T_4$ (500 ppm) followed by $T_3$ (395 ppm), $T_2$ (380 ppm), $T_1$ (334 ppm) and $T_0$ (292 ppm) at the end of composting. Ihsanullah Daur [67] suggested that the mineralization of organic waste resulted in a decrease in mass and an increase in micronutrients. The consortium of *Eisenia fetida* and *Trichoderma harzianum* led to the higher micronutrients in $T_4$ because *Eisenia fetida* excreted mucopolysaccharides, which aerated the soil and provided optimum conditions for the growth of fungus, leading to proper degradation of rice straw. Lakshmi et al. [52] reported an increase in Fe by 77.5%, Zn by

93.3% and Cu by 166.67% in a vermicompost mixture of different straw wastes. A similar trend of increased Zn and Fe was observed in rice straw compost by Jusoh et al. [68]. The results are in accordance with the earlier studies of composting [6,30].

## 4. Conclusions

It is evident from the results of this study that composting paddy straw using lignocel-lulolytic fungi and earthworms in combination could be a valuable strategy to increase the proportion of the final compost product. The accelerated mineralization rate yielded many-fold increases in levels of essential plant nutrients (e.g., N.P.K). The decreased C:N ratio of the final compost showed a favorable compost with good stability and maturity. This study demonstrated that inoculation with *Eisenia fetida*, *Trichoderma harzianum* and cow dung converts agricultural waste into a value-added product, i.e., the compost having a better degree of maturity. The novelty of this study was to assess and compare the utilization of a lignocellulolytic fungus alone and in combination with earthworms to improve compost maturation. Statistical analysis by two-way ANOVA suggested that the combinatorial approach for composting affected the mineralization rate significantly. This study provides a solid base to study the utilization of compost inoculation in real conditions on agricultural fields as an alternate technology for managing paddy straw.

**Author Contributions:** N.S.: conceptualization, investigation, writing the original draft. J.S.: visualization, data curation, statistical analysis. B.S.: methodology and design of study, revision of manuscript. V.M.: conception, validation, supervision. All authors have read and agreed to the published version of the manuscript.

**Funding:** This research was financially supported by the Council of Scientific and Industrial Research (CSIR), New Delhi (Award no. 09/382(0225)/2019-EMR-I).

**Institutional Review Board Statement:** Not applicable.

**Informed Consent Statement:** Not applicable.

**Data Availability Statement:** All the data associated with this article are within the article.

**Acknowledgments:** The authors are thankful to the IARI, New Delhi, for providing *Trichoderma harzianum* (ITCC 6721) and *Eisenia fetida* for experimental work.

**Conflicts of Interest:** The authors declare no conflict of interest.

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
