# Peer review of "Improving the Agronomic Value of Paddy Straw Using Trichoderma harzianum, Eisenia fetida and Cow Dung"

_fermentation, doi:10.3390/fermentation9070671_

Round 1

Reviewer 1 Report

TITLE -  refers to the content of the manuscript.

INTRODUCTION - is  a proper introduction to the topic of work; it does  clearly specify the subject matter.

MATERIALS AND METHODS - has little novelty and is already used in similar investigations; information on the population of Eisenia fetida is missing.

RESULTS AND DISCUSSION - are satisfactory; the description of figures is clear and understandable.

CONCLUSIONS -  are general; the innovativeness of the conducted research was not emphasized; the conclusion section lacks a perspective for future research; can be improved.

REFERENCES - are adequate, correct and in line with the previous chapters. The use of the bibliography is correct

Recommendation

I recommend this manuscript for publication in the journal Fermentation with minor corrections.

Author Response

We are highly thankful to the Reviewer for critical examination of our manuscript and providing useful comments. All changes are marked in red colour in revised manuscript. Our responses are given below:

Reviewer 1:

Comment 1: Material and Methods - has little novelty and is already used in similar investigations; information on the population of Eisenia fetida is missing.

Revision: Material and methods has been revised by adding relevant references and information about population of Eisenia fetida has been provided.

Comment 2: Conclusions - are general; the innovativeness of the conducted research was not emphasized; the conclusion section lacks a perspective for future research; can be improved.

Revision: conclusion has been rewritten in the light of emphasizing the present study’s results, with defining innovativeness and future perspectives.

Reviewer 2 Report

The manuscript looks enough good, but:

1. The first paragraph of "3 Results and discussion" must be moved to introduction, as there are no results from your study.

2. The suggestion for improvement would be to include additional statistical analysis to support the results. More detailed information on the statistical tests employed and the significance of the observed results are needed. Results do not have statistical evaluation.

3. The main shortcoming of results is that subchapters from 3.1 to 3.10 are written in the same structure. It starts from describing min/max values, then in which order are differences, then finds some reference for similar results. It is very boring to read the same texts just with different numbers. I suggest to rewrite the whole part of results - from 3.1 to 3.10 subchapters.

There are some minor errors.

Author Response

We are highly thankful to the Reviewer for critical examination of our manuscript and providing useful comments. All changes are marked in red colour in revised manuscript. Our responses are given below:

Reviewer 2:

Comment 1: The first paragraph of “3 Results and discussion” must be moved to the introduction, as there are no results from your study.

Revision: The first paragraph of the results and discussion has been deleted and rewritten with a concise description of the results.

Comment 2: The suggestion for improvement would be to include additional statistical analysis to support the results. More detailed information on the statistical tests employed and the significance of the observed results are needed. Results do not have statistical evaluation.

Revision: Statistical analysis has been added along with statistical evaluation and the significance of results has been provided.

Comment 3: The main shortcoming of results is that subchapters from 3.1 to 3.10 are written in the same structure. It starts from describing min/max values, then in which order are differences, then finds some reference for similar results. It is very boring to read the same texts just with different numbers. I suggest to rewrite the whole part of results - from 3.1 to 3.10 subchapters.

Revision: All the subchapters from 3.1-3.10 has rewritten as per suggestion.

It is further stated that the manuscript has been checked with the premium version of Grammarly to remove grammatical errors and improving English in general.

Round 2

Reviewer 2 Report

The paper was corrected.